# The Role of Procalcitonin as an Antimicrobial Stewardship Tool in Patients Hospitalized with Seasonal Influenza

**DOI:** 10.3390/antibiotics12030573

**Published:** 2023-03-14

**Authors:** Ingrid Christensen, Dag Berild, Jørgen Vildershøj Bjørnholt, Lars-Petter Jelsness-Jørgensen, Sara Molvig Debes, Jon Birger Haug

**Affiliations:** 1Østfold Hospital Trust, Department of Infection Control, 1714 Kalnes, Norway; lars.p.jelsness-jorgensen@hiof.no (L.-P.J.-J.); jon.birger.haug@so-hf.no (J.B.H.); 2PhD Program Medicine and Health Sciences, Faculty of Medicine, University of Oslo, 1072 Oslo, Norway; 3Institute of Clinical Medicine and Department of Infectious Diseases, Oslo University Hospital, 0450 Oslo, Norway; dag.berild@medisin.uio.no; 4Institute of Clinical Medicine, Faculty of Medicine, Oslo University, 0450 Oslo, Norway; joerbj@ous-hf.no; 5Department of Microbiology, Oslo University Hospital, 0372 Oslo, Norway; 6Faculty of Health and Social Studies, Østfold University College, 1671 Fredrikstad, Norway; 7Østfold Hospital Trust, Center for Laboratory Medicine, 1714 Kalnes, Norway; sara.molvig.debes@so-hf.no

**Keywords:** antimicrobial stewardship, antimicrobial resistance, seasonal influenza, procalcitonin, hospital, diagnostic tool

## Abstract

Background: Up to 60% of the antibiotics prescribed to patients hospitalized with seasonal influenza are unnecessary. Procalcitonin (PCT) has the potential as an antimicrobial stewardship program (ASP) tool because it can differentiate between viral and bacterial etiology. We aimed to explore the role of PCT as an ASP tool in hospitalized seasonal influenza patients. Methods: We prospectively included 116 adults with seasonal influenza from two influenza seasons, 2018–2020. All data was obtained from a single clinical setting and analyzed by descriptive statistics and regression models. Results: In regression analyses, we found a positive association of PCT with 30 days mortality and the amount of antibiotics used. Influenza diagnosis was associated with less antibiotic use if the PCT value was low. Patients with a low initial PCT (<0.25 µg/L) had fewer hospital and intensive care unit (ICU) days and fewer positive chest X-rays. PCT had a negative predictive value of 94% for ICU care stay, 98% for 30 days mortality, and 88% for bacterial coinfection. Conclusion: PCT can be a safe rule-out test for bacterial coinfection. Routine PCT use in seasonal influenza patients with an uncertain clinical picture, and rapid influenza PCR testing, may be efficient as ASP tools.

## 1. Introduction

Respiratory tract infections (RTI) are the leading cause of hospital antibiotic prescriptions and a significant issue in the medical field, leading to increased antibiotic resistance and higher healthcare costs. One reason for excessive antibiotic use is the difficulty differentiating between a severe viral infection and bacterial pneumonia [1,2]. Since the etiology is often viral, antibiotics will, to a large extent, not benefit these patients. A large meta-analysis found bacterial coinfection in only one-fourth of patients with influenza [3]. Recent rapid bedside diagnostics of influenza and other viral respiratory agents have improved the opportunity to reduce antibiotic misuse. Nevertheless, 60–80% of patients with influenza-like illnesses are still prescribed antibiotics [4,5]. Thus, there is a need for improved antimicrobial stewardship (ASP) efforts for RTI and a potential regarding ASP by more efficiently using diagnostic markers.

The serum inflammatory marker procalcitonin (PCT) is downregulated in viral and upregulated in bacterial infections. It increases and decreases faster in the course of infection than the more routinely used inflammatory marker C-reactive protein (CRP) [6]. The test can consequently help to differentiate between viral and bacterial infections and monitor infection progress. Several studies have shown the usefulness of PCT in reducing antibiotic use, predominantly in sepsis and lower respiratory tract infections, and international consensus guidelines have recommended PCT as a decision aid in ASPs [7,8,9]. Yet, controversy exists about PCTs’ ASP attributes because high-quality studies have reported neutral or negative results for the usefulness of PCT [10,11,12]. This controversy partly explains why PCT is not systematically used in ASPs, even in high-income countries, as exemplified by a hospital survey recently published in Norway [13].

A recent search of Medline and Embase revealed 34 studies on the clinical utility of PCT in hospitalized patients with influenza. However, in the two systematic reviews retrieved, only H1N1 influenza patients from the 2009 pandemic were included [14,15] (Search string found in Appendix A). In addition, seasonal influenza differs from pandemic influenza in the immunity response [16]. Therefore, the role of PCT as an ASP tool in patients with seasonal influenza remains unclear. 

We aimed to explore the role of PCT as an ASP tool in patients hospitalized with seasonal influenza by investigating any association between PCT levels and the use of antibiotics, the need for intensive care, and mortality at 30 days.

## 2. Results

### 2.1. Patients

We included 116 (20%) out of 589 patients hospitalized with influenza at Østfold Hospital Trust during the 2018–20 influenza season. No patient declined participation, but we excluded two patients due to a PCT measurement exceeding 48 h from inclusion. The remaining 473 influenza patients were not available for inclusion because the microbiologists who reported positive results (patients) to us could only prioritize this extra work when they had time and not at the expense of their regular duties.

Eighteen (16%) of 116 patients had influenza as the tentative diagnosis at admission, while 34 (29%) had a bacterial infection as the tentative diagnosis. For a complete overview of admission diagnoses, see Appendix A. Few patients had their PCT measurements on day three (N = 27) and day five (N = 32). The reason was either because no blood samples were taken or the patient had been discharged. Therefore, analyses are based on the admission PCT value (N = 116) because of the small samples for days three and five and because initial PCT measurements were deemed more realistic in easy-to-implement future ASPs. 

Patient characteristics and main findings are given in Table 1. The CRB-65 score estimates the severity of community-acquired pneumonia based on the four criteria: confusion (yes/no), respiratory rate ≥ 30 (yes/no), blood pressure (systolic < 90 mmHg or diastolic ≤ 60 mmHg), and age ≥ 65, Based on the score, the patients are ranked into strata of low, moderate, or high risk of mortality [17]. 

The highest proportion of patients stayed in the infection and pulmonary wards, while a low proportion of the included patients stayed in the intensive care unit (ICU). 

The median value of the inflammatory markers leucocytes and PCT were within the reference ranges, while CRP was above but in the lower part of the CRP scale. The high proportion of Influenza A compared to Influenza B was representative of previous influenza seasons in hospital surveillance.

### 2.2. Primary Outcome

#### Antibiotic Use

In the linear regression analyses, after adjusting for age, sex, and other confounders (Appendix A), we found that a higher level of PCT at admission was significantly associated with increased use of antibiotics, measured in DDD (defined daily doses) and DOT (days of therapy) (Table 2).

At the time of inclusion, 86 (98%) patients were prescribed antibiotics (i.e., after being diagnosed with influenza). Patients receiving antibiotic treatment at any time during the hospital stay were 88 (76%). Within three days after the diagnosis, 20 patients (23%) had their antibiotics de-escalated (stopped or changed to a narrower spectrum), and their median admission PCT level was 0.1 µg/L (IQR 0.05–0.65). Sixty-six patients (77%) continued the antibiotic regime, and their median admission PCT value was 0.32 µg/L (IQR 0.12–1.6). Thus, we found a significant difference in the admission PCT levels between the patients who had their antibiotics de-escalated and those who did not (*p*-value 0.04). 

### 2.3. Secondary Outcomes

#### Mortality and Intensive Care Stay

In the logistic regression analyses (Appendix A), we found that the odds ratio for PCT as the predictor of 30 days mortality (outcome) was 1.2 (95% CI 1–1.4), which means that for a one µg/L increase in PCT, the odds of 30 days mortality increased by 20% and thus had a significant association with PCT (*p*-value 0.023). There was, however, no significant association between PCT and the need for intensive care during the hospital stay (Table 2). 

Six (5.2%) patients died in the hospital. Causes of death, registered in the patients’ journal, were pneumonia (three patients), respiratory failure (one patient), pulmonary edema (one patient), and organ failure (one patient). Nine patients (7.7%) died within 30 days post-discharge. Causes of 30 days mortality were not registered.

### 2.4. In-Between Group Differences

From the in-between group differences, we could infer that those patients with low PCT values (<0.25 µg/L) stayed significantly fewer days in the hospital and in the intensive care unit, had lower CRP, less positive chest X-rays, were prescribed lesser amounts of antibiotics, which, in a larger proportion, were narrow-spectrum. Moreover, we found a significantly higher 30 days mortality in the patients with high PCT values (≥0.25 µg/L). (Table 3).

### 2.5. Bacterial Coinfection and Microbiology

Seventeen (15%) of 116 patients had a bacterial coinfection confirmed by clinically significant positive sputum samples, blood culture results, and/or urinary antigen tests (UAT). One of the 105 patients tested had a positive blood culture isolate (*Staphylococcus aureus).* Eleven patients of 67 tested (16%) had positive sputum culture (three *Moraxella catarrhalis*, four *Haemophilus influenzae*, three *Streptococcus pneumoniae*, and one isolate with both *H.influenzae* and *S.pneumoniae*). Seven patients of 46 tested (15%) had positive UAT for *Streptococcus pneumoniae*. Two patients had more than one positive culture and were counted only once. The patients with bacterial coinfection had a median PCT of 0.5 µg/L (IQR 0.05–1.64) versus those without confirmed bacterial coinfection with a median PCT of 0.17 µg/L (IQR 0.07–0.7; *p*-value 0.6).

Table 4 presents the test performance, using 0.25 µg/L as the cut-off, based on previous literature [9]. PCT has a high sensitivity and a high negative predictive value (NPV) for an ICU stay and 30 days mortality. With an area under the receiver operator curve (AUROC) of 0.73 for ICU stay and 0.8 for 30 days mortality; thus, the overall test performance is good. ROC curves of PCT on bacterial coinfection, ICU stay and 30 days mortality are provided in Figure 1. However, PCT has a low specificity and positive predictive value (PPV) for the same outcomes (ICU stay and 30 days mortality), thus, the test performance is better for predicting a true negative outcome than for predicting true positive outcomes. For predicting bacterial co-infection, PCT has a sufficient NPV of 88% but a low sensitivity and poor overall test performance with an AUROC of 0.51.

## 3. Discussion

Our main findings were that higher PCT levels in hospitalized influenza patients were significantly associated with increased antibiotic use and 30 days mortality. Furthermore, PCT had a high negative predictive value (NPV) for a stay in the ICU and mortality. This is not to say that PCT can be used to predict (or negatively predict) either mortality or an ICU stay. However, the findings suggest that using a low PCT cut-off (<0.25 µg/L) as a rule-out tool for antibiotic treatment in hospitalized influenza patients can be safe in terms of clinical deterioration. While interpreting these findings, it is worth noting that as an ASP tool, PCT may only supplement the clinical presentation, which is the main determinant for any antibiotic decision [18]. 

Furthermore, in our study, all the patients had influenza (as an inclusion criterion). There were significantly lower admission PCT levels in patients who had their antibiotics stopped or de-escalated than in those who continued their antibiotics unchanged. A previous study found that early influenza diagnoses decreased antibiotic prescribing and, thus, had an essential role in ASP [1]. Therefore, adding PCT in this setting may strengthen the ASP aspect. 

We also found that the PCT value did not differ significantly between patients with or without bacterial coinfection, and the positive predictive value (PPV) for bacterial coinfection was only 20%. Thus, the diagnostic ability (rule-in value) of PCT was insufficient. However, the NPV of PCT for bacterial coinfection was 88%. Therefore, PCT has more potential as a bacterial coinfection rule-out test in our study, a finding in line with updated PCT guidelines [9,19]. A systematic review and an observational study of 972 patients concluded that PCT was a suitable rule-out test but not for detecting (ruling in) bacterial coinfection [15,20]. Another systematic review, however, found PCT to be an accurate marker for detecting bacterial pneumonia [14]. However, caution must be taken comparing these previous studies with the research presented here since they either included a much higher proportion of ICU patients [15] or exclusively used ICU patients [14,20]. Moreover, the studies reported mainly on 2009 pandemic influenza patients, which might not be transferable to seasonal influenza due to differences in the pathogenesis [16].

Significantly more patients with a high PCT value had a chest X-ray infiltrate than those with a low PCT. This indicates that a positive chest X-ray may add crucial clues for suspecting a bacterial coinfection, although from only a radiological view, the differentiation is difficult [21]. Nevertheless, the value of chest X-ray is readily available, considering it is a routine procedure in most Norwegian hospitals for most admitted somatic patients.

Most of our influenza patients had a low CRB-65 score, and only 15% had a bacterial coinfection; still, 76% were prescribed antibiotics. Previous studies on pandemic influenza patients showed a higher proportion of bacterial coinfection of 21–50%, which may indicate an underreporting of bacterial infections in our population [14,15,22]. However, natural fluctuations in small samples of patients may partly explain this difference. Moreover, most studies were investigating ICU patients, while in our study, the participants were from all departments, i.e., our population was principally less sick. Nonetheless, our finding of the overuse of antibiotics supports the call for antibiotic stewardship in influenza patients.

We observed that few prescribing physicians ordered PCT routinely. Consequently, the PCT results were, in most cases, unavailable on admission for the physician. Instead, the first author (IC) ordered PCT after inclusion, i.e., when influenza was confirmed. At this point, the physician had already given a (tentative) diagnosis and initiated treatment. As our study observations are from a “real-world” scenario, we find the relatively low compliance of physicians with established PCT algorithms to be a valuable observation. Previous literature shows that PCT algorithm adherence is inconsistent between 44% and 100% [11,23]. One often used explanation for low to moderate PCT adherence is differences in study designs and lack of real-world insights [11,23,24]. A recent qualitative study from our institution found that physicians were uncertain of PCT’s correct use and trustworthiness [18]. Given our observation of low PCT algorithm adherence and PCT’s still unleashed potential in an ASP setting, we propose that further efforts should focus on the education of prescribing physicians to achieve an earlier and more consistent use of PCT during the influenza season.

Our study has some limitations. First, one should be careful to generalize our findings based on results from a single center and a relatively small number of patients. We included only 20% of the total number of influenza patients hospitalized over two winter seasons, and our results may, therefore, not be representative of the total influenza population. However, a selection bias is less likely since the inclusion of patients was dependent solely on the capacity of the microbiologists who reported the PCR-positive influenza tests to the study group, i.e., the microbiologists did not see the patients, only their positive PCR test. 

Second, the low antimicrobial resistance (AMR) in Norway might reduce the external validity of this study [25]. Finally, one might postulate a lesser risk perception of antibiotic treatment failure in Norwegian physicians as a reason for not using PCT routinely. However, non-compliance with PCT algorithms in the hospital care setting with high pressure on caregivers has been discussed in several studies from countries with considerably higher AMR [26].

Our study also has strengths. To our knowledge, the study is the first to report procalcitonin’s role in antibiotic use in exclusively seasonal influenza patients. By describing real-life data, the study adds implications for future ASPs. Our conclusions of the need for reinforced use of PCT and focus on early influenza diagnosis are also reasonably easy to implement, with potentially large rewards in ASP given the up to 60% excess antibiotic use in influenza patients [3]. Based on the study findings and the body of evidence, although subtle regarding seasonal influenza, we suggest physicians can safely supplement their clinical evaluation with a PCT test in seasonal influenza patients. Furthermore, we advocate for further exploration of PCTs’ potential as an AMS tool, preferably intervention studies.

## 4. Materials and Methods

### 4.1. Setting 

This prospective, single-center observational study was performed at Østfold Hospital Trust (a secondary, acute care, 380-bed hospital in southeastern Norway) with a catchment area of 320,000 inhabitants. In the study hospital, an ASP was established in 2017. We did no systematic interventions as part of this study. Only regular activities by the antibiotic stewardship team took place during the study period, such as attendance at clinical visits with information on the utility of PCT and distribution of posters with PCT algorithms.

### 4.2. Data Collection

We included patients 18 years and older from February to April 2019 and October 2019 to March 2020. The inclusion criteria were a positive polymerase chain reaction (PCR) for influenza and a PCT assay obtained less than two days after the influenza diagnosis. When the microbiologist had the capacity, they called the first author (IC) or the last author (JBH) and reported the positive PCR results for seasonal influenza. IC or a research fellow contacted the patient (the same day or the day after) to inform them about the research and allowed them to opt-out. For included patients, the attending physicians or IC ordered serial measurements of PCT on days 1, 3, and 5 (±two days). However, PCT was only measured if routine test panel was drawn so that no extra procedure was needed for the patients. 

In addition to PCT, the following variables were registered: patient age and gender, total days in the hospital, influenza type and other respiratory viruses (respiratory syncytial virus, human metapneumovirus, parainfluenza- and adenovirus), results of positive blood cultures, sputum, and urinary antigen test (UAT) for *Streptococcus pneumoniae*, admission diagnosis, total leukocyte count, and CRP on days 1, 3 and 5 (±two days), in hospital and 30 days case fatality, the department where the patient had the most protracted stay, admittance in the intensive care unit, days of symptoms before PCR result, presence of chest X-ray infiltrates, CRB-65 score [17] (scored based on information in the patient journal). Furthermore, antibiotic treatment before hospital admission, as well as antibiotics administered on hospital days 1, 3, and 5, were registered and categorized as narrow- or broad-spectrum antibiotic regimes. All penicillins without inhibitory enzymes, gentamicin, and co-trimoxazole were denounced as narrow-spectrum antibiotics. Cephalosporins (except first Generation), quinolones, penicillins with inhibitory enzymes, carbapenems, erythromycin, doxycycline, metronidazole, and vancomycin were categorized as being broad-spectrum [27]. Finally, we measured the administered amount of antibiotics in defined daily doses (DDD) and the number of days of antibiotic therapy (DOT) for each patient.

### 4.3. Statistical Analysis

We divided the patients into one low (<0.25 µg/L) and one high PCT (≥0.25 µg/L) group to compare in-between differences and used the Wilcoxon rank sum test for continuous and Fisher exact test for categorical variables to test significance. *P*-values less than 0.05 were considered statistically significant. We used the Shapiro–Wilk test for normality testing; none of the variables were normally distributed. Categorical variables are presented in absolute values, and percentages and continuous variables are presented as medians and interquartile range (IQR).

We used multivariate logistic regression models to investigate procalcitonin levels associated with the following outcome variables: need for a stay in the ICU and 30 days mortality. We used multivariate linear regression to analyze hospital antibiotic use as the outcome, measured in defined daily doses (DDD) per 100 patient days and days of therapy (DOT). In the linear regression model, we did a logarithmic transformation in the variables that did not have a normal distribution or a linear relationship. In all regression models, we adjusted for confounders and omitted colliders [28] (Appendix A).

We chose the 30 days over the in-hospital mortality rate in the analyses, as it is a commonly used quality indicator [29]. To calculate PCT’s sensitivity and specificity values, we used 0.25 µg/L as the cut-off [9].

STATA version 15.1 (StataCorp LP, College Station, TX, USA) was used for all statistical analyses. Appendix A contains all statistical analyses, including the variables we adjusted for.

## 5. Conclusions

We have demonstrated that a procalcitonin measurement in the context of influenza diagnosis has the potential as a safe rule-out test for bacterial coinfection and that the diagnosis of influenza itself probably has a role in reducing antibiotic use. Implementing routine PCT use in patients admitted to hospitals with seasonal influenza with an uncertain clinical picture has potential rewards regarding ASP, particularly when combined with rapid influenza tests. We suggest that physicians may use PCT as a clinical adjunct in seasonal influenza patients, ideally by implementing unambiguous PCT algorithms. Furthermore, we call for intervention studies to further quantify PCTs’ impact as an ASP tool in this patient group.

## Figures and Tables

**Figure 1 antibiotics-12-00573-f001:**
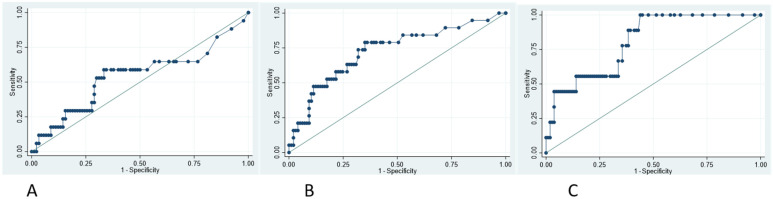
Receiving operator curve of PCT for (**A**): bacterial coinfection, (**B**): intensive care unit stay, and (**C**): 30 days mortality.

**Table 1 antibiotics-12-00573-t001:** Patient characteristics in hospital.

	Proportion (Number of Patients)
**Median age 71 years (range 22–96)**	100% (116)
Female/male	41%/59% (47/69)
CRB-65 score 0–1	62.9% (73)
CRB-65 score 2	22.4% (26)
CRB-65 score 3–4	13.8% (16)
Prehospital antibiotics	8.6% (10)
Influenza type A/B	96.5%/3.5% (112/4)
Respiratory syncytial virus/human metapneumovirus	N = 1/1
**Wards (longest stay)**	
Infection	33% (38)
Pulmonary	22.4% (26)
Geriatric	11.2% (13)
Oncology	7.7% (9)
Other (cardiology, surgery, nephrology, observation, neurology, hematology)	19.8% (23)
Intensive care unit	6% (7)
Intensive care unit during hospital stay	16.4% (19)
**Inflammatory laboratory tests**	**Median value (min–max; IQR)**
CRP day 1 (mg/L)	65 (1–323; 29–112)
PCT day 1 (µg/L)	0.16 (0.02–28.5; 0.07–0.71)
Leucocytes day 1 (10^9^/L)	5.5 (0.6–16.8; 4.6–7.5)
**Days in hospital**	4 (1, 34; 2–6.5)

CRB-65 score: a pneumonia severity score based on Confusion (yes/no), Respiratory rate ≥ 30 (yes/no), blood pressure (systolic < 90 mmHg or diastolic ≤ 60 mmHg), and age ≥ 65, CRP = C-reactive protein, PCT = Procalcitonin, IQR = interquartile range.

**Table 2 antibiotics-12-00573-t002:** The association of procalcitonin with the outcomes: antibiotic use, mortality, and a stay in the ICU.

Outcome Variable: Days of Antibiotic Therapy ^1^	β-Coefficient (95%CI)	*p*-Value
Procalcitonin	0.12 (0.01–0.23)	**0.026**
Outcome variable: Antibiotic therapy in DDD/100 patient days ^1^	**β-coefficient (95%CI)**	
Procalcitonin	0.17 (0.04–0.3)	**0.007**
Outcome variable: 30 days mortality ^2^	**OR (95% CI)**	
Procalcitonin	1.2 (1–1.4)	**0.023**
Outcome variable: Intensive care unit stay ^2^	**OR (95% CI)**	
Procalcitonin	1.1 (1–1.2)	0.14

^1^ Linear regression model. The variables that were not normally distributed are logarithmic transformed. ^2^ Logistic regression model; DDD = Defined daily doses, OR = Odds ratio, CI = Confidence interval.

**Table 3 antibiotics-12-00573-t003:** Comparison of low and high PCT patients.

	Procalcitonin Level Day 1		
	<0.25 µg/L (N = 64) (Low)	≥0.25 µg/L (N = 52) (High)	
**Patients’ characteristics**			*p*-value *
Age (range; IQR)	68.2 (22–95; 53.7–78.8)	74.3 (26–96; 64.3–81)	0.08
	Median (IQR)		
Days in hospital	3 (2–5)	5 (3–9)	**<0.001**
Days with symptoms before the result of virus PCR was obtained	3 (2–7)	4 (2–7)	0.39
Leucocytes on admission	5.2 (4–7.2)	6.1 (4.3–7.7)	0.43
CRP on admission	40 (15.5–75)	116 (64, 173)	**<0.001**
	N (% of total patients)		*p*-value
Bacterial coinfection **	7 (6%)	10 (8%)	0.43
CRB-65 score < 2 on admission	45 (39%)	28 (24%)	0.06
30 days mortality	1 (0.9%)	8 (6.9%)	**0.01**
Intensive care stay	4 (3.4%)	15 (13%)	**<0.01**
Positive chest X-ray	14 (12%)	29 (25%)	**<0.01**
**Antibiotic use**	Median (interquartile range)		*p*-value
Days of antibiotic therapy (DOT)	2 (0–4.5)	4 (2–7.5)	**<0.01**
Antibiotic therapy in DDD/100 PD	5 (2.7–8.2)	8.4 (4–17.7)	**<0.01**
	N (% of the 86 patients receiving antibiotics at admission)		*p*-value
Narrow-spectrum antibiotics (only)	26 (30%)	18 (20%)	**0.03**
Broad-spectrum antibiotics used	15 (17%)	27 (31%)	**0.03**

* Wilcoxon rank sum for continuous variables and Fisher’s exact test for categorical variables; ** Positive blood culture, sputum or urinary antigen test (UAT), alone or in combination; IQR = interquartile range, CRP = C-reactive protein, PCT = procalcitonin, DDD = Defined daily doses, PD = patient days.

**Table 4 antibiotics-12-00573-t004:** Performance of the procalcitonin test in hospitalized patients with seasonal influenza.

	Sensitivity	Specificity	PPV *	NPV *	AUROC ᵜ
**Bacterial coinfection** **	59%	54%	20%	88%	0.51
**Intensive care stay**	79%	62%	29%	94%	0.73
**30 days mortality**	88.9%	58.9%	15.4%	98%	0.8

* PPV and NPV: positive and negative predictive value; ** Bacterial coinfection confirmed by positive sputum, blood culture, and/or UAT; ᵜ AUROC: area under the receiving operator curve.

## Data Availability

The data presented in this study are available on request from the corresponding author. The data are not publicly available because the patients were not informed about this.

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
