# Peer review of "The Role of Procalcitonin as an Antimicrobial Stewardship Tool in Patients Hospitalized with Seasonal Influenza"

_antibiotics, 2023, doi:10.3390/antibiotics12030573_

Round 1

Reviewer 1 Report

Dear Author,

Please see my comments below and in the attached file,

Kind regards,

My comments:

Title: do you mean reduce antibiotics' "use" or "need"? If yes, please amend the title accordingly. The title is somehow confusing. please can consider rewriting the title more clearly and properly to fit with the aim of the study.

Introduction:

The introduction is short and needs to be expanded.

line 59: In the abstract, you said the  "role" and here you said "use-fullness"? Please be consistent.

Results:

Table 1 line 71: " IQR" Abbreviation not defined before?

do you mean the interquartile range?

Discussion:

Line 157-158: "Caution must be taken in 157 the comparison of these studies with ours, since they included a higher proportion of ICU 158 patients [15] and only ICU patients [14, 20]." this is a confusing statement. please consider rewriting.

Line 181,182,183: Do any other studies have the same result (i.e., low physician compliance)? please compare with the results from other studies in the literature.

Line 188: Your study is a "single"-center study. please address this as a limitation.

line 205: Please add a paragraph here where you clearly describe the impact of your study finding on healthcare professionals and policymakers (i.e., what is/are the key messages based on your study finding to the healthcare providers and policymakers?).

Materials and methods:

Setting: line 208: Your study is a "single"-center study. please address this as a limitation in the discussion section. please refer to my comments there.

Conclusion:

This section is weak. please consider rewriting this section as below:

first, start by stating the main finding, then describe the impact of your main finding, and finally, any future recommendations? any future studies?

Reviewer 2 Report

The manuscript entitled “Can Procalcitonin reduce antibiotics in patients hospitalized with seasonal influenza?” is interesting, relevant, and well written and should be included in the special issue “Antibiotic Use and Stewardship in Hospital”.

The study was well conducted, the results are adequately treated, and they are presented and discussed in a clear, logical and coherent way.

Thus, the paper should be accepted in present form.

The only 2 changes I suggest are: (1) in line 82 change "..., and 86 (98%) patients..." to "..., and 86 (98%) of those patients..."; and (2) the title of the subsection "In-between group differences" (line 105) should be placed to the next page.

Reviewer 3 Report

This is an interesting subject with interesting results.  However, in several places phrasing makes your intended meaning confusing.  The reason for using IQR for age is also not clear.  

Below are suggested changes preceded by line #

19

All data was obtained from a single  clinical setting and analyzed by descriptive statistics and regression models. 

64 

We included a total of 116 (20%) out of the 589 patients hospitalized with influenza at Name of Hospital during the 2018–20 influenza seasons at .

70

Table 1. elaborate on the Table data.  Explain CRB-65 score system in a sentence or 2.  Note differences in flu type.  Comment on Wards and lab tests.

71

Patient characteristics.

Table 1

Move column subheadings down to a row, changing first cell from “Age” to Patients (age range) and 2nd cell to 116

move “Proportion (Number of patients)” to title row

Put a line break between each subsection (for example, before “Wards (longest stay)

Add “Laboratory tests” or something similar above the cell “CRP day 1 (mg/L)”

79

clarify when PCT measurements were taken into relation of when antibiotic therapy was started.  This statement makes it sound like you are suggesting antibiotic use may have caused the high PCT levels.

82

…, and 86 (98%) of those patients were prescribed….

84-86

clarify when PCT levels were measured.  Did the 20 de-escalated patients have 0.1ug/L PCT at 3 days, before antibiotics were stopped?  The 66 patients who stayed on antibiotics have their PCT tested at d3?  Please rewrite these sentences to clarify  this point. 

92

…increased 20% in patients with PCT values greater than 1 ug/L….

92

“(95% CI 1–1.2)”  according to the table 1-1.2 is the range for the ICU patients

93

this doesn’t seem to be the case.  ICU patients have nearly the same PCT as 20-day mortality.  Why is the p value so different?  

100-101

move this sentence to M&M’

Table 2.

Column 1 looks off

“Outcome variable: Days of antibiotic therapy”  should be moved down one row.

106 

there is a big difference in age between the 2 groups.  statistical significance should consider the true range and not just IQR

112

Table 3. Comparison of low and high PCT patients 

125-125

For PCT 0.5 include units

You report IQR for one group and range for another, this is confusing.  Why is a p value of 0.6 reported here, is comparing the two groups? No significance?  Please clarify

128

“Table 4. shows a high negative predictive value and low positive predictive value of…”  Join this sentence with the above paragraph

130

explain sensitivity, specificity, PPV and NPV outcomes in more detail.  Explain AUROC and why it is listed here.

135-136

This one sentence suggests that high PCT levels have PPV and NPV for mortality.  Also, table 3 shows that high PCT patients stayed in the ICU more than low.  How is it a NPV indicator?

144-146

again, the timing between antibiotic use, stoppage and PCT measurements needs to be clarified.  

156-157

Another systematic review, however, found PCT to be an accurate marker for the detection of bacterial pneumonia [14]. Caution must be taken comparing these previous studies with the research presented here since they either included a much higher proportion of ICU patients [15] or exclusively used ICU patients [14, 20].

179-180

what is the average time from antibiotic treatment, influenza diagnosis and PCT data

208 

…was performed at Name of Hospital (a secondary, 208 acute care, 380-bed hospital in the southeastern part of Norway) with a catchment area of 320,000 inhabitants.

Reviewer 4 Report

I have read the manuscript titled "Can Procalcitonin reduce antibiotics in patients hospitalized with seasonal influenza?"

I have some comments and queries before consideration for publication.

Title:

·        It is not advisable to have a question as the title of your paper.

Introduction:

·        Line 53: A recent search (06.1.23) of Medline and Embase >> either remove the date since it is mentioned in the Supplement file or write it in the full form to avoid confusion.

·        Line 57: the role of PCT as an antimicrobial stewardship (ASP) >> no need to mention the long form of the abbreviation ASP as it was mentioned before in line 50.

Results:

·        Line 64: We included a total of 116 (20%) out of 589 patients hospitalized with influenza >> please mention the reasons for excluding other patients (473) as it was mentioned that “No patient declined participation”.

·        Table 1: CRB-65 score 0–1 62.9% (73) >> most of the patients had a low CRB-65 score and could have been treated as outpatients. Were there any other indications for admissions other than the severity of the respiratory infections?

·        Table 1: Although presenting data as Median value (min-max; IQR) is more comprehensive, I recommend presenting CRP, PCT, and Leucocytes values in table 1 as Median (IQR).

·        Line 82: and 86 (98%) patients were prescribed antibiotics at the time of inclusion >> better rephrase to avoid confusion about the percentage e.g., and 86 (98%) of them were prescribed antibiotics.

·        Line 92: by 30 days post-discharge increased by 20% (95% CI 1–1.2) for a one μg/L increase in PCT >> in table 2 CI is 1–1.4, please correct.

·        Line 125: a median PCT 0.5 (IQR 0.5–1.64) >> median = Q1, please correct.

·        Line 126: a median PCT of 0.17 (range 0.07–0.7; p-value 0.6) >> is it range or IQR, please clarify.

·        Table 4: it is advisable to present ROC figures if available.

Discussion:

·        Line 135-137: Our main findings were that higher PCT levels were significantly associated with increased antibiotic use and mortality and PCT had a high negative predictive value (NPV) for a stay in the ICU and mortality. >> please rephrase to be more specific that your findings were in hospitalized influenza patients and 30-day mortality.

·        Lines 138-139: However, the findings suggest that using a low PCT cut-off (< 0.25 μg/L) as a rule-out tool for antibiotic treatment can be safe in terms of clinical deterioration. >> please rephrase to be more specific that your findings were in hospitalized influenza patients.

·        Lines 163-164: This indicate that a positive chest X-ray may add crucial clues for suspecting a bacterial coinfection, although from a radiological view the differentiation is difficult [21]. >> The authors cited a website (Radiopaedia). However, I suggest citing journal articles e.g., DOI: https://doi.org/10.1016/j.ejrad.2004.03.010. Furthermore, X-rays can identify other pathologies e.g., pneumothorax and pleural effusion which may need immediate interventions.

·        Line 188: Our study has some limitations >> the authors should mention among the limitations the single-center observational nature of the study.

·        Line 194: Second, the low antimicrobial resistance (AMR) in Norway might reduce the external validity of this study. >> what is the reference for antimicrobial resistance in Norway?

Materials and Methods:

·        Line 218: When they had the capacity, a microbiologist called the first author (IC) or the last author (JBH) and reported the positive PCR results for seasonal influenza. >> it is unclear on what base the authors include their patients. What do they mean by “When they had the capacity”?

·        Line 221: the attending physicians or IC ordered serial measurements of PCT on days 1, 3, and 5 (+/- two days). >> although the authors state that they did a serial PCT measurement, it was not clear throughout the manuscript if their findings were related to admission PCT or serial PCT (this applies to the whole results and discussion sections).

·        Lines 233-236: All penicillins without inhibitory enzymes, gentamicin, and co-trimoxazole were denounced as narrow-spectrum antibiotics. Cephalosporins (except 1. generation), quinolones, penicillins with inhibitory enzymes, carbapenems, erythromycin, doxycycline, metronidazole, and vancomycin were categorized as being broad-spectrum. >> please mention the base or reference of this categorization. Please change 1. generation to first generation.

Conclusions:

·        Line 262: We have demonstrated that procalcitonin has the potential as a safe rule-out test >> please specify which PCT; admission or serial? 
